# Study of Atmospheric Forcing Influence on Harbour Water Renewal

Yaiza Samper *, Manuel Espino *, Maria Liste, Marc Mestres, José M. Alsina and Agustín Sánchez-Arcilla

Laboratori d'Enginyeria Marítima (LIM), Universitat Politècnica de Catalunya (UPC-BarcelonaTech), C/Jordi Girona, 08034 Barcelona, Spain; maria.liste@upc.edu (M.L.); marc.mestres@upc.edu (M.M.); jose.alsina@upc.edu (J.M.A.); agustin.arcilla@upc.edu (A.S.-A.)
* Correspondence: yaiza.julia.samper@upc.edu (Y.S.); manuel.espino@upc.edu (M.E.)

**Abstract:** In this study, we use observations and numerical simulations to investigate the effect of meteorological parameters such as wind and atmospheric pressure on harbour water exchanges. The modelled information is obtained from the SAMOA (Sistema de Apoyo Meteorológico y Oceanográfico de la Autoridad Portuaria) forecasting system, which is a high-resolution numerical model for coastal and port-scale forecasting. Based on the observations, six events with high renewal times have been proposed for analysis using the SAMOA model. Therefore, the conclusions of this study have been possible due to the combination of observed data from the measurement campaigns and the information provided by the model. The results show that days with higher renewal times coincide with favourable wind-direction events or increases in atmospheric pressure. After analysing these events using model results, it was observed that during these episodes, water inflows were generated, and in some cases, there was a negative difference in levels between inside and outside the harbour produced by atmospheric pressure variations. The latter may be due to the fact that the water in the harbour (having a lower volume) descends faster and, therefore, generates a difference in level between the exterior and the interior and, consequently, inflow currents that imply an increase in the renewal time. These results are a demonstration of how meteorological information (normally available in ports) can be used to estimate currents and water exchanges between ports and their outer harbour area.

**Keywords:** water exchanges; harbour renewal time; wind and atmospheric pressure effects; Huelva; Gijón and Cartagena harbours

## 1. Introduction

Coastal zones are important for human activity worldwide. They are areas of important production centres, human settlements and tourist destinations, and consequently, areas with high population densities. The European Commission estimates that the population in European coastal regions is around 194 million people, 38% of the total population [1]. Coastal cities and port activity both have a strong influence on water quality and renovation inside harbours and surrounding areas, contributing to the deterioration of coastal ecosystems, loss of biodiversity and destruction of habitats. At the same time, water circulation in coastal areas is affected by climatic events that can be extreme, such as storms and strong winds [2], which influence the port-external water exchanges and water renovation. In order to preserve the environmental health of coastal zones and improve their protection, it is important to establish and use long-term environmental management tools. European legislation is the main guideline for developing these tools and as well as to comply with international standards [3]. Understanding and being able to predict the physical behaviour of coastal and harbour zones is a key tool to mitigate and reduce anthropogenic impacts resulting from the exploitation of these areas.

According to the survey carried out in 26 European harbours in the framework of the PEARL project (Port EnvironmentAl infoRmation colLector), the main environmental

interests in European harbours are those related to the monitoring of currents, tides and waves, which is helpful in ensuring the safety of navigation, predicting the dispersion of pollution, and identifying sources of pollution, etc. The survey also reveals the importance of water quality, meteorological parameters, sedimentation and turbidity processes [3]. In order to respond to these needs and assess the actual state and dynamics of the seas, as well as to obtain forecasts at different spatial and temporal scales, significant progress has been made in recent years in the implementation of operational ocean forecasting systems [4]. The Copernicus Marine Environmental Monitoring Service (CMEMS) reflects significant advances in operational oceanography. This service provides regular and systematic information on the physical state of the global ocean and European seas [5]. All CMEMS-derived products depend on in situ and satellite observations, which are used to develop new products and validate models [6]. The SAMOA (Meteorological and Oceanographic Support System of the Port Authority) forecast systems are CMEMS downstream services, being the coastal models nested into the regional CMEMS Iberia Biscay Irish forecast solution (CMEMS IBI) [7]. One of the main goals of SAMOA is to provide a series of high-resolution numerical models for the prediction of ocean-meteorological variables and forecasting inside of harbours, with coastal and harbour scales.

High-resolution forecasting models are used to support different activities in harbour domains, for example, hydrodynamic simulation to help the handling of large vessels; to anticipate harbour closure due to extreme events; to propose improvements in operations and safety; as a source of information for the design of contingency plans in case of spills; to propose environmental management plans; and to comply with current legislation related with water quality, etc. The availability and development of more advanced forecasting systems with different domains and scales have made it possible to use the models in a wide range of areas. The most frequent areas of application are those related to the simulation of accidental spills [8]; the physical impacts of storms and climate change [9–11]; the study of dominating harbours hydrodynamics [12,13]; the influence of tides on port water quality [14]; and the effects of port activity on nearby urban areas [15]. These forecasting systems make it possible to study currents in different situations and under different environmental conditions. It is a useful tool to analyse port hydrodynamics in strong wind conditions, without wind, during storms or under static atmospheric conditions; on the other hand, it is useful to analyse the influence of external currents on the internal domain and to estimate how this affects the renewal capacity and, therefore, the quality of the water. Usually, areas with low water turnover (high turnover time) are associated with a high risk of water quality degradation [16]. This information can be used to develop plans to minimise the risk of future accidents.

The water quality of a harbour and its ecological status are mainly determined by its capacity for renewal and mixing. The renewal capacity of water in semi-enclosed domains (like ports) is mainly conditioned by the exchange with the outer domain but also by the internal circulation within the harbour [12,17]. This water exchange can be influenced by multiple meteo-oceanographic [18], geometric and geographic conditions [19,20]. Due to their complex geometry and infrastructure (e.g., quays, channels and docks), circulation within ports tends to be low, which enhances the stagnation and confinement of pollutants [21]. The physical characteristics of the harbours and the region are static and invariable, but the weather conditions are constantly changing and play an important role in circulation. These, at the same time, may have a physical impact on the harbour's infrastructure. Therefore, accurate model predictions of water inflow and outflow under different meteorological conditions are desired for water quality monitoring. Reliable and verified information on meteo-oceanographic dynamics can improve residence time estimation and improve environmental management systems.

Some studies consider meteorological parameters to characterise the hydrodynamics or water quality of beaches [22], bays [23] or harbours [13,24]. The innovation, in this case, is that the analysis is focused on their effect on the water exchanges between the inside and outside of several harbours. The central aim of the study is to analyse the

influence of atmospheric forcings on water renewal rates in harbour domains. Specifically, to examine the effect of weather conditions on water inflows and outflows in ports based on observations and modelled data. In other words, to relate the variability of the renewal time of each harbour to meteorological weather, in particular wind and atmospheric pressure.

This article is organised as follows: first, after this brief contextualisation, the Section 2 describes the study areas that are the focus of this work and the measurement campaigns. Then, the SAMOA forecasting system and the validation methodology are presented. The Section 3 is organised in two parts: the first is based on observations, and the second is based on model results. In the first part, one describes the meteorology and hydrodynamics of each port during the measurement campaigns and highlights events of interest with high renewal times. In the second part, a selection of these events is presented in 2D images from the SAMOA forecasting model. Fourthly, the Section 4 provides an analysis of the results obtained and presents different meteorological scenarios that may justify the renewal times observed above. Finally, the paper concludes with some suggestions and proposals for future studies.

## 2. Materials and Methods

This section describes the procedure followed to derive water quality information from observational data and numerical simulations. First, for a correct interpretation of the data, the ports and the field campaigns are described. Then, the SAMOA forecasting model is explained. To finish, the validation of the model is presented.

### 2.1. Study Area

This work focuses on the ports of Huelva, Gijón and Cartagena (southwest, north and southeast of the Iberian Peninsula). These harbours are located in different seas and have different physical characteristics (see Figure 1). A study area has been delimited in each port (where the acoustic doppler current profilers-ADCP were located). Table 1 shows the main geometrical characteristics of these domains.

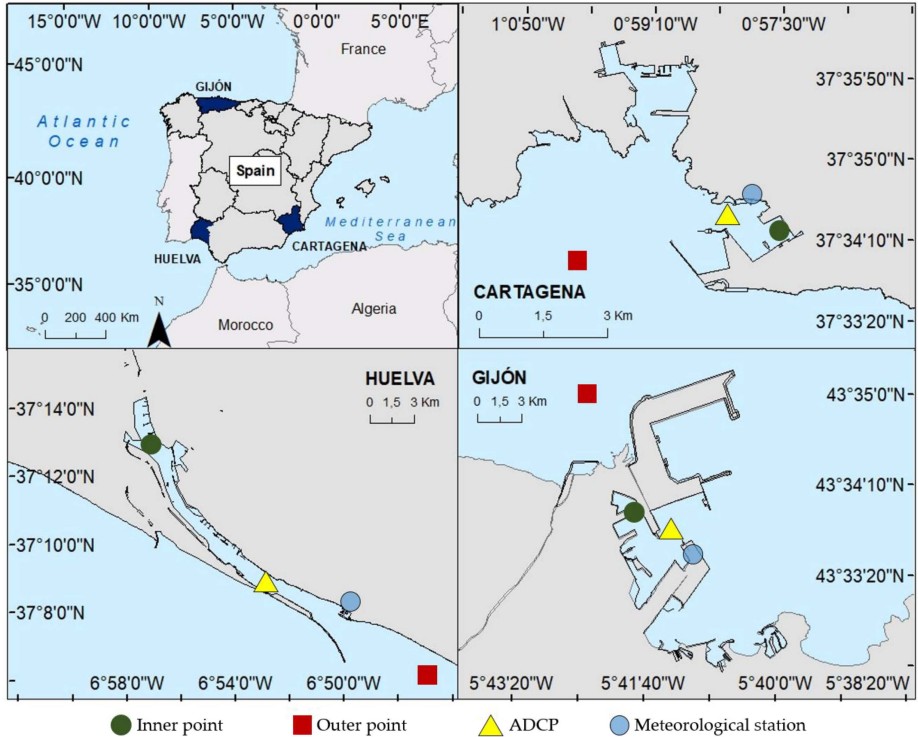

**Figure 1.** Location of the ports studied. The red square corresponds to the outer points discussed below; the green pin corresponds to the inner points; the yellow triangle represents the ADCP locations, and the blue circle indicates the meteorological stations.

**Table 1.** Main characteristics of each study area.

| Harbour | Surface (km$^2$) | Mean Depth (m) | Volume Considered (m$^3$) |
|---------|------------------|-----------------|----------------------------|
| Huelva | 5.04 | 10.3 | 51.91200 |
| Gijón | 0.95 | 11.4 | 10.80162 |
| Cartagena | 0.95 | 20.3 | 19.31455 |

### 2.1.1. Huelva Harbour

The port of Huelva is located in the Odiel estuary, in the southwest of the Iberian Peninsula. This tidal region is connected to the Atlantic Ocean and has tides of up to 3.84 m [25], which are responsible for the exchanges between the water masses circulating within the estuary, determining the dilution capacity in the estuarine system [26]. On a large scale, this region is located in the Gulf of Cadiz, whose circulation is determined by the two-layer water exchange between the Atlantic and Mediterranean basins through the Strait of Gibraltar [27,28]. Its location allows the port to receive daily inflows of water from both the ocean and the rivers Tinto and Odiel, which flow into the estuary. The port has the shape of a channel, and it is 13 km long and has depths between 500 and 1000 m. Its bathymetry is between 5 and 10 m in the inner area and reaches 20 m in the outer area of the port (Figure 2a).

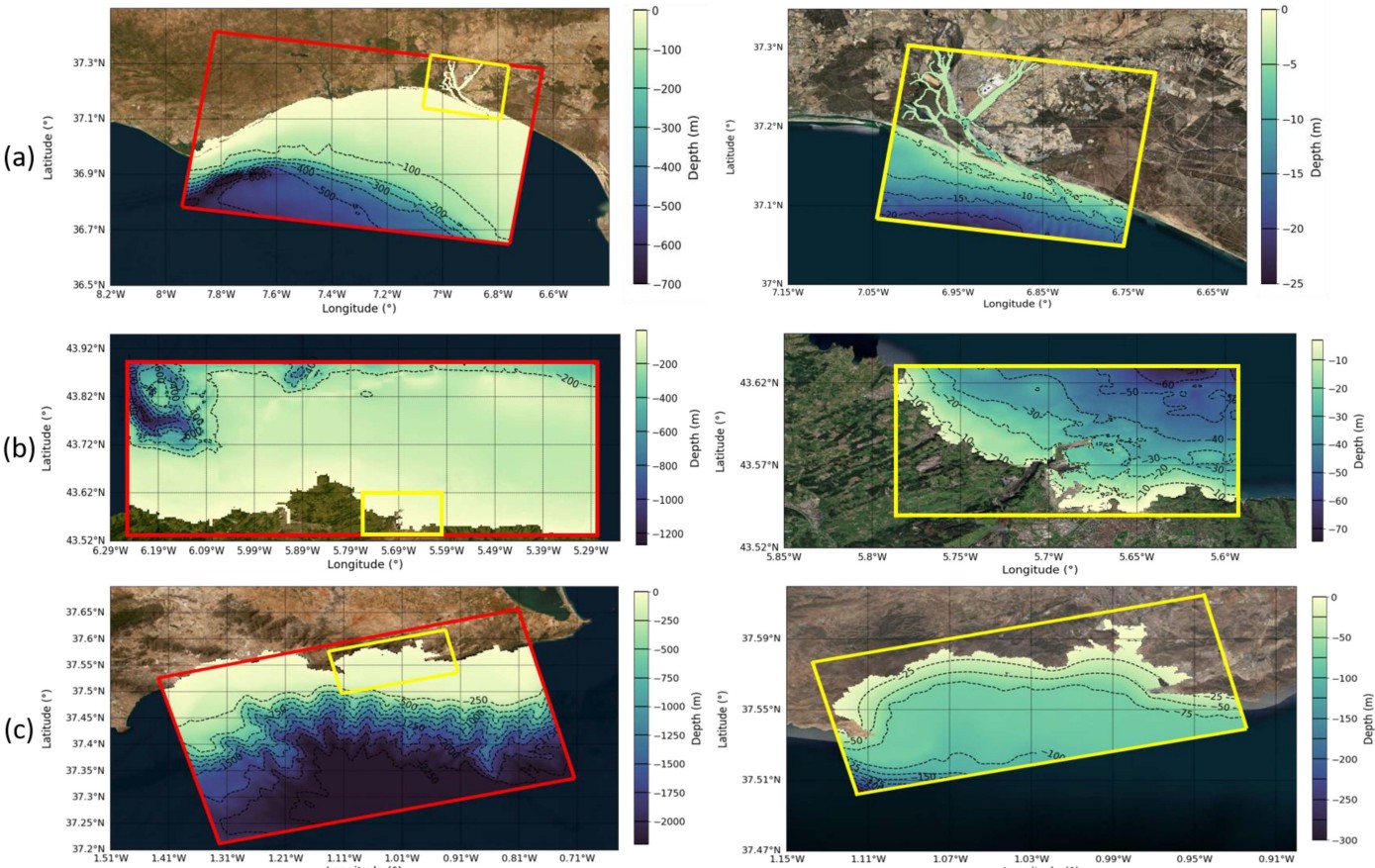

**Figure 2.** Location and extension of the coastal (red box) and harbour (yellow box) domains in Huelva (**a**), Gijón (**b**) and Cartagena (**c**). The bathymetry of each domain is also shown (yellow corresponds to shallower and dark blue to deeper areas).

### 2.1.2. Gijón Harbour

The port of Gijón is located in the southern Gulf of Biscay (Cantabrian Sea), in the north of the Iberian Peninsula and has tides of up to 4.62 m [29]. The ocean circulation in this region shows complex patterns and currents varying in intensity and direction

depending on the time of year. It is also strongly influenced by the Gulf Stream and the complex marine topography [30,31]. The port is divided into three independent areas: Puerto del Musel, Puerto Deportivo and Marina Yates. The area around the harbour has a complex topography, dominated mainly by the *Peñas* cape and the *Amosucas* shelf (not visible in Figure 2b). These geographical features provide natural protection against storms and modify the incoming waves by diffraction and refraction [2]. The bathymetry of the inner harbour shows depths of 5 to 20 m (Figure 2b).

### 2.1.3. Cartagena Harbour

Cartagena Port is located in the Bay of Cartagena, in the southeast of the Iberian Peninsula. This bay is in the Mediterranean Sea (semi-enclosed sea) and therefore does not have significant tides (maximum tidal range 0.95 m [32]). This Port has an "open" shape; it has no defined channel and is formed by two independent basins: The Cartagena basin and the Escombreras basin. The maximum depth in the harbour is 25 m and can reach 75 m in the outermost area (Figure 2c).

### *2.2. Fiel Campaigns*

Observations were collected in the framework of the SAMOA project from 12 October 2020 to 8 February 2021 in Cartagena, from 14 April to 13 July 2021 in Huelva, and from 10 November 2021 to 20 January 2022 in Gijón. Time series of currents, sea level and meteorological data were measured in the three harbours.

Water currents measurements were obtained by an acoustic doppler current metre (ADCP) in each harbour (AWAC 1MHz model NORTEK-AS). Both have been manufactured by Nortek (Rud, Norway). In Huelva, the ADCP was located inside the estuary where the port begins. Given the complex geometry and the absence of other measuring devices, this work will focus on the first 6 km of the estuary. In Gijón Harbour, it was located at the mouth of the inner Musel Harbour basin, and this work will focus only on this domain. In Cartagena, it was located at the Escombreras basin entrance, and this work will be focused on this basin.

Meteorological and sea level data were recorded by two meteorological stations with a tide gauge in each harbour. One station was located near the ADCP, while the other was outside the port. All the time series data has a sampling interval of 10 min.

For reading, representation and analysis of time series data (wind, level, currents and temperature), Python programming language was used. In particular, for the graphical representation, Matplotlib and Pandas libraries were used, and a 12-h moving average for a better graphic display is generally applied.

### *2.3. Renewal Times*

The methodology used to calculate the renewal time follows the techniques described in [20], which assumes that water exchanges take place only at the mouth of the port, ignoring the permeability of the dikes and overtopping. In this study, to estimate the renewal time in the ports of Huelva, Gijón and Cartagena, the second method presented in the mentioned article was used. That is, based on the measurements of currents at the mouth and the dimensions of each study area, the inflow and outflow of water and the renewal time (TR) were calculated. In particular, TR was obtained by dividing the volume of each port by the mean daily outflows, as in [20]. The procedure for the calculation of TR used is as follows:

Outflow rate (m$^3$/s) = outflow mean velocity (m/s) x outflow mouth area (m$^2$)

TR = Port volume (m$^3$)/outflow rate (m$^3$/s)

### *2.4. The SAMOA Operational Service*

SAMOA forecasts were generated using the Regional Ocean Modelling System (ROMS). This model has been developed by (Dr. Hernan G. Arango, in Institute of Marine and

Coastal Sciences Cook Campus, Rutgers University in New Jersey, USA; and Dr. Alexander F. Shchepetkin in Institute of Geophysics and Planetary Physics in California, USA). This system covers different basins at regional scales (resolution varying from 1.8 to 2.2 km) [33] and uses different forcing and nesting strategies [34], [35]. Numerical details are described in Shchepetkin and McWilliams [33], and complete information on the model and its source code are available on the ROMS website (http://myroms.org/ accessed on 15 January 2023).

The SAMOA initiative is the solution of Puertos del Estado (PdE) and the Spanish port authorities to respond to the needs of metocean information at coastal and port scales [34]. This service includes a high-resolution operational system in domains such as ports and coastal areas.

The SAMOA model application consists of two nested regular grids with a spatial resolution of ~350 m and ~70 m for the coastal and harbour domains, respectively. The vertical discretisation consists of 20 sigma levels in the coastal domains and 15 levels for all port domains [36]. In this study, the results of the harbour grid for Gijón and Cartagena and of the coastal grid for Huelva have been used. The reason for not using the harbour grid domain in Huelva (with a larger resolution) is that it was not yet operational during the field campaign period. The characteristics of the computational domains are summarised in Table 2.

**Table 2.** Characteristics of the computational domains.

| Harbour | Domain | Extension (km) | Dimension (Cells) |
|---|---|---|---|
| Huelva | Coastal | 106 × 70.5 | 303 × 202 |
| | Harbour | 26 × 24.3 | 372 × 347 |
| Gijón | Coastal | 77.9 × 40 | 223 × 115 |
| | Harbour | 15.5 × 9.9 | 222 × 142 |
| Cartagena | Coastal | 55.4 × 39 | 159 × 112 |
| | Harbour | 17.26 × 8.13 | 177 × 132 |

To provide sufficiently detailed bathymetry, the SAMOA model uses a combination of global (GEBCO-https://www.gebco.net/, accessed on 20 January 2023) and local data sources (provided by the port authorities). Figure 2 shows the location and extension of the coastal (red box) and harbour (yellow box) domains in all the ports studied, with the corresponding bathymetry in contour plots ranging from yellow to blue (from shallowest to deepest).

The SAMOA models are nested into the daily regional forecasts delivered by CMEMS-IBI (Copernicus Marine Environment Monitoring Service-Iberia Biscay Irish) [4]. At the sea surface, the SAMOA models are forced by high frequency (hourly) wind stress, joint with atmospheric pressure, fluxes of water (evaporation minus precipitation) and surface heat derived from the Spanish Meteorological Agency forecast (AEMET) (based on the AEMET HARMIONE model 2.5 km application nested into the ECMWF IFS forecast). CMEMS-IBI provides hourly data for water currents and sea levels and are applied as open boundary conditions (OBC). Moreover, it also provides daily values of temperature and salinity in the water column. Where freshwater discharges may be relevant, the river discharge is taken into account, considering climatological data and a constant salinity of 18 PSU. A detailed description of the SAMOA operational system can be found at 35].

### 2.5. Validation

The numerical model validation was performed by comparing observations with the mean numerical result of the four points in the numerical grid close to the observation location. For the present study, model results for atmospheric pressure, meteorological tide and currents were used; therefore, the qualitative and quantitative validation of these variables was carried out. Other studies, such as [35–37], used the model in different conditions and environments. For this validation, the number of data values presented in

Table 3 have been used, which corresponds to hourly data during the whole duration of the campaigns.

**Table 3.** Number of data (observations and model results) used for the validation of the variables analysed.

| Variable | Number of Data for the Validation | | |
|---|---|---|---|
| | **Huelva** | **Gijón** | **Cartagena** |
| Atmospheric pressure | 1729 | 224 | 452 |
| Meteorological tide | 1793 | 1463 | 2752 |
| Currents through the mouth | 2161 | 1729 | 2752 |

The validation process was carried out in two phases: an initial plotting of time series at hourly intervals; and a second analysis by calculating different basic statistics such as BIAS, RMSE and correlations (R and values of the correlation line). Before starting the validation process, the amount of data from the observations and the model results were harmonised because the campaign data were 10 minutely and the model data were hourly.

Qualitative analysis of the atmospheric pressure, meteorological tidal, and currents time series (an example is presented in Figure 3) shows the SAMOA model results are in accordance with the observations in all three harbours. This validation is completed in the Supplementary Information document (Figure S1. Time series and scatter plots of model results and observations).

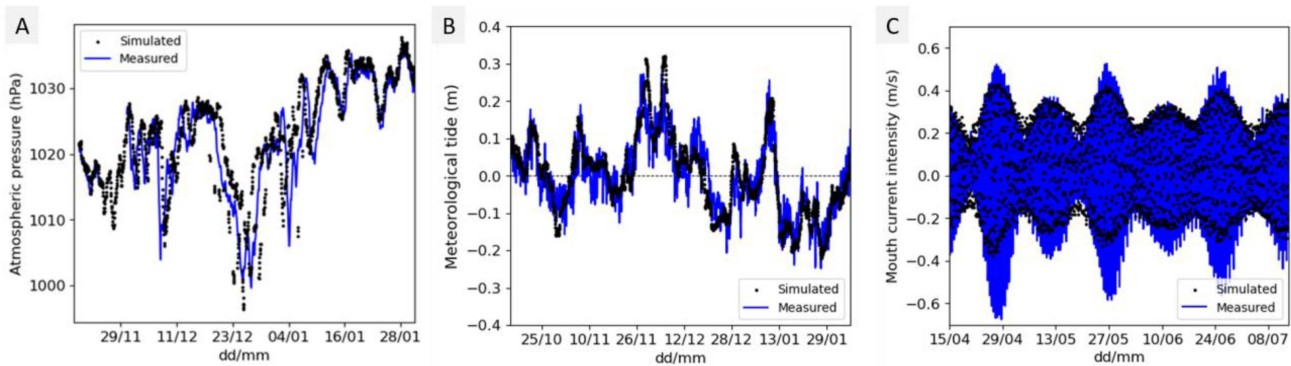

**Figure 3.** Time series of atmospheric pressure in Gijón (**A**), meteorological tide in Cartagena (**B**) and currents through the mouth in Huelva (**C**). The observations correspond with the blue lines and forecasted by SAMOA are the black dots.

The statistical analysis (presented in Table 4) confirmed the fit of the model for the observations, with correlations of 0.89, 0.57 and 0.76 for Huelva; 0.99, 0.76, and 0.29 for Gijón; and 0.99, 0.88 and 0.34 for Cartagena respectively. It should be noted that both the measurements and the model prediction are conditioned by the complexity of each domain, the harbour infrastructures and the intensity of the tides. The correlation coefficients between the model results and the observations are relatively low in some parameters, but it has been noted that they reproduce the observed trends; that is, if the observations show inflows or rises in sea level, the model (with slightly different values) does reproduce these inflows or rises.

**Table 4.** Model validation metrics for simulated atmospheric pressure, meteorological tide and currents through the mouth compared with observation from campaign data.

| Variable | Statistical | Huelva | Gijón | Cartagena |
|---|---|---|---|---|
| Atmospheric pressure | RMSD | 256.23 hPa | 120.38 hPa | 111.17 hPa |
| | R | 0.61 | 0.87 | 0.83 |

**Table 4.** *Cont.*

| Variable | Statistical | Huelva | Gijón | Cartagena |
|:---:|:---:|:---:|:---:|:---:|
| Meteorological tide | RMSD | −0.0009 m | −0.01 m | 0 m |
|  | R | 0.57 | 0.73 | 0.88 |
| Currents through the mouth | RMSD | −0.01 m/s | −0.03 m/s | 0.009 m/s |
|  | R | 0.76 | 0.45 | 0.34 |

## 3. Results

In the three harbours analysed, the time series of renewal time shows days with some high values (Figures 4a, 5 and 6a). Table 5 shows the average renewal time calculated from the currents' measurements at each harbour mouth. These results are 27, 21 and 25 days for the ports of Huelva, Gijón and Cartagena, respectively. The days above average are 28%, 27% and 25% of the total series in each case. In order to understand the origin of these above-average TR values, wind and atmospheric pressure variations (from campaign data), together with currents and sea level (from observations and model results), were studied. To facilitate the analysis of the current behaviour and the distribution of the renewal times during the periods studied, histograms with the frequency of the TRs in each case are presented in the Supplementary Information document (Figure S2. Histogram of renewal time distribution in the ports of Huelva, Gijón and Cartagena).

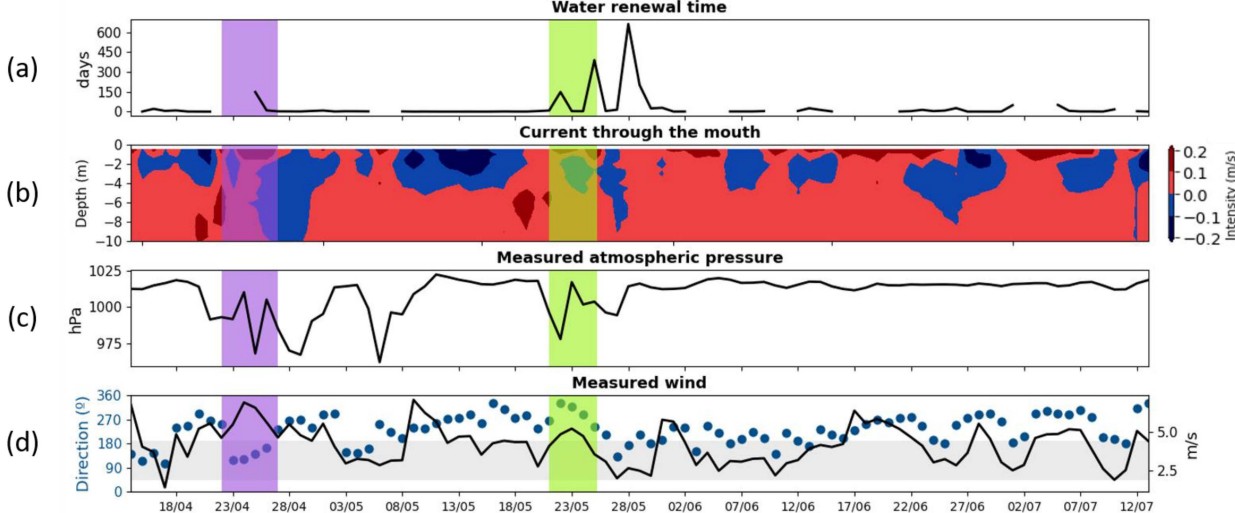

**Figure 4.** From top to bottom, water renewal time (**a**), currents at the mouth (**b**), measured atmospheric pressure (**c**), and measured wind (**d**) (the grey shaded area highlights the direction that allows water inflow) in Huelva's harbour. The boxes show two examples of high renewal time episodes: the purple one, related to an episode of wind favourable to water inflow, and the green one, linked to an increase in atmospheric pressure.

**Table 5.** Average renewal time (days) calculated from the data obtained during the measurement campaigns and days (%) above this average of the total campaign data.

|  | Mean TR | Days above Average (%) |
|:---:|:---:|:---:|
| HUELVA | 27 days | 28 |
| GIJÓN | 21 days | 27 |
| CARTAGENA | 25 days | 25 |

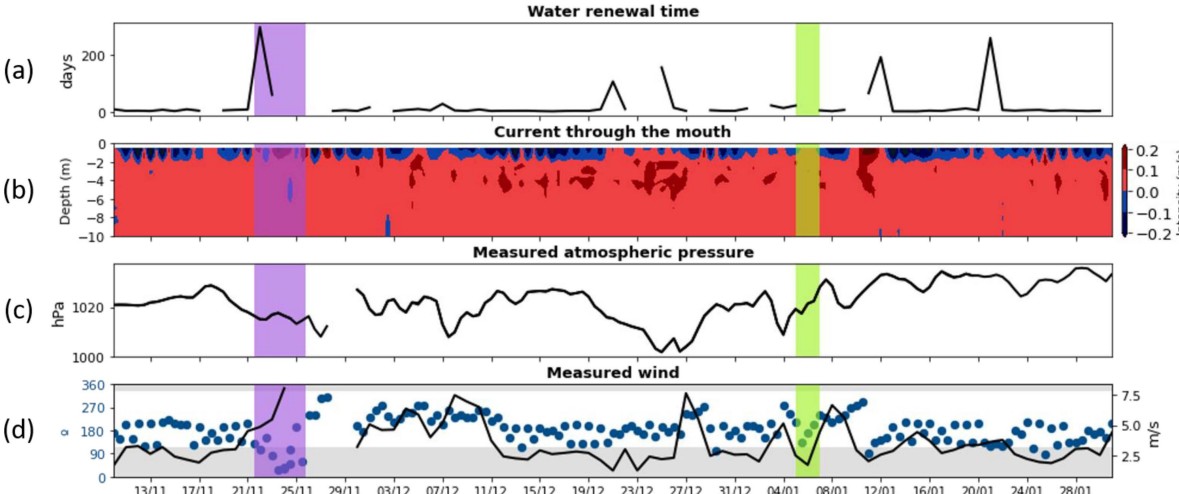

**Figure 5.** From top to bottom, water renewal time (**a**), currents at the mouth (**b**), measured atmospheric pressure (**c**), and measured wind (**d**) (the grey shaded area highlights the direction that allows water inflow) in Gijón's harbour. The boxes show two examples of high renewal time episodes: the purple one, related to an episode of wind favourable to water inflow, and the green one, linked to an increase in atmospheric pressure.

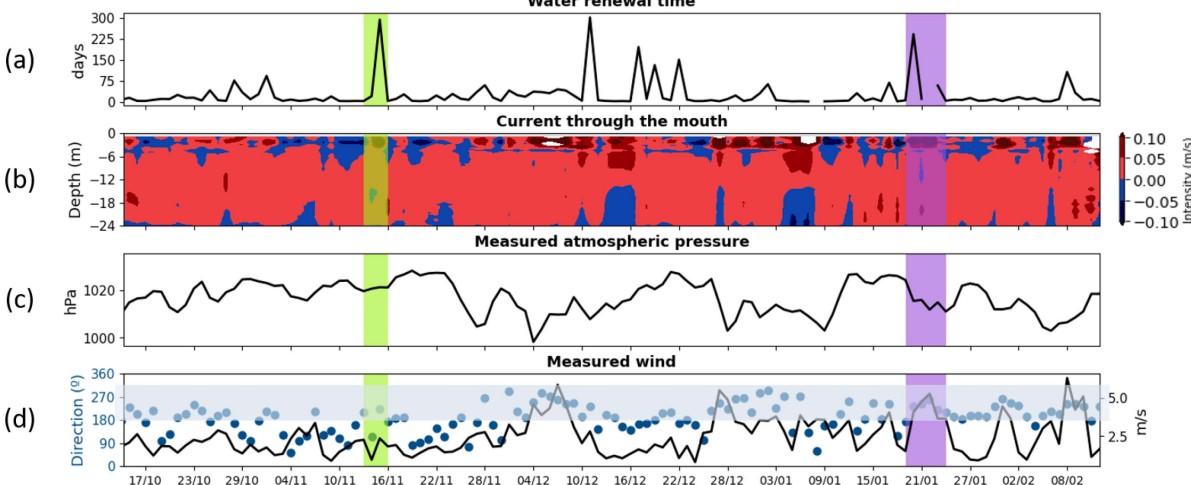

**Figure 6.** From top to bottom, water renewal time (**a**), currents at the mouth (**b**), measured atmospheric pressure (**c**), and measured wind (**d**) (the grey shaded area highlights the direction that allows water inflow) on Cartagena's harbour. The boxes show two examples of high renewal time episodes: the purple one, related to an episode of wind favourable to water inflow, and the green one, linked to an increase in atmospheric pressure.

### 3.1. Metocean Field Data

The renewal time series (Figures 4a, 5 and 6a) show data gaps on some days. These data gaps correspond to days during which there are no outflows of water; that is, the outflow is almost zero for the whole day. High renewal times, therefore, indicate that there is more water coming in than going out. Figures 4a, 5 and 6a show the harbour mouth current intensity (positive values signify inflow and are in red).

To identify the possible reason for these high inflows, the winds during these events were analysed. It was observed that some of these TR (Renewal time) peaks coincided with wind episodes (vertical purple shades areas in Figures 4a, 5 and 6a) that favoured water inflow (these favourable directions are highlighted by a grey horizontal band). However, others coincided with positive gradients in atmospheric pressure (vertical green shades areas in Figures 4a, 5 and 6a). In other words, some TR peaks (defined as peaks with

above-average values) appear to be related to wind events favourable to water inflow and others to increases in atmospheric pressure.

In the case of the port of Huelva, observing the time series in Figure 4, there are 25 days with TR above the average. These high TRs seem to be related, on some occasions, to winds that favour the entry of water; and, on others, to increases in atmospheric pressure. In this figure, one event of each type (wind with a purple band and one atmospheric pressure with a green band) has been selected for later model analysis.

In the port of Gijón, 23 days were observed with renewal times (or data gaps implying an outflow close to 0 m$^3$/s) above average. During some of these events, the wind favoured the entry of water into the port and during others, the atmospheric pressure increased (in Figure 5, one of these wind events is highlighted with a purple band and one atmospheric pressure with a green band). In the following section, these events will be analysed on the basis of the results provided by the model.

In the Cartagena harbour, 30 days with renewal times above the mean (or data gaps) have been detected. In the course of these events, winds that favour the entry of water into the port and atmospheric pressure increases happen. In the following section, two of these events are highlighted in Figure 6 and will be analysed using the results provided by the model.

After calculating the time series of renewal time for each harbour, the average renewal time for the period has been calculated (Table 5), and the days with the TR above the average have been selected. For each individual case, wind and atmospheric pressure have been studied, obtaining the results presented in Table 6 (a recount of the days with peaks that were apparently related to the entry of water due to a favourable wind direction; the days with peaks that seem to be related to the entry of water due the increase in atmospheric pressure; and the days with high values that do not seem to be related to either of these two causes).

**Table 6.** Summary of the causes that have been identified from the measured time series to justify the increases in renovation times in the three ports analysed.

| | Days with Renewal Time Above Average | | |
|---|---|---|---|
| | Cause | | |
| | Wind (%) | Atmospheric Pressure (%) | Unknown (%) |
| HUELVA | 20 | 32 | 48 |
| GIJÓN | 26 | 44 | 30 |
| CARTAGENA | 53 | 27 | 20 |

*3.2. Model Results*

After analysing the observations (which correspond to a single measuring station), the model results were used to complement them and to study the currents in and around the harbour more extensively. This section examines the inflow and outflow currents and sea level variations during six of the identified events with high renewal times, two for each harbour.

The results provided by the model indicate that, during these events, water inflows into the harbour predominate, which agrees with observations. In the following, high TR events associated with wind events favourable to water inflow and with episodes of increased atmospheric pressure will be analysed. Therefore, modelled results of two events will be presented for each harbour (the first related to wind and the second to atmospheric pressure).

Figure 7 shows the circulation pattern in the Port of Huelva and its coastal area corresponding to the results provided by the model for 24 April 2021. According to the observations, during this day, there was an episode of intense wind from the southeast which could favour the entry of water into the port through its mouth oriented in this same

direction. The model results show inflow currents for the same day, which would justify the increase in TR.

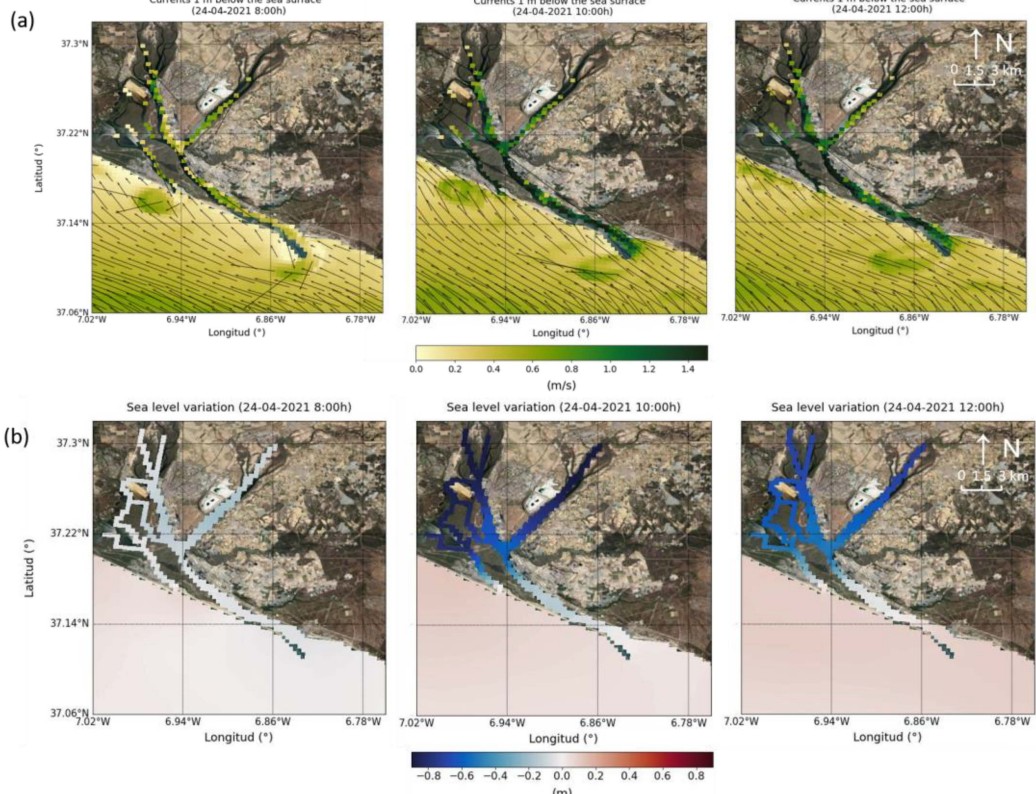

**Figure 7.** Intensity and direction of the sea currents (**a**) and sea level variation (**b**) in the port of Huelva during the southeast wind episode on 24 April 2021.

Figure 8 presents the sea level variation in the port of Huelva based on the results provided by the model for 22 May 2021. During the first hours of the day, the figure shows an irregular sea level variation for the area inside and outside the harbour. In the inner harbour, this variation is negative, i.e., the level inside the harbour is falling, reaching 80 cm below the mean level. However, the outer harbour area maintains a constant level without significant changes.

Figure 9 presents the currents in the port of Gijón and its coastal area based on the model results for 23 November 2021. According to the observations, during this day, there was an episode of strong north-easterly wind, which seems to have favoured the entry of water into the harbour through its mouth. The model shows inflow currents, which could justify the TR peak on this day.

Figure 10 shows the sea level variation in the port of Gijón from the results provided by the model for 4 January 2022. In the figure, it can be seen that the variation in sea level is not equally distributed between the harbour area and the area outside the harbour. In the sheltered area of the harbour, this variation is negative, i.e., the level is decreasing, reaching 2 cm below the mean level. However, in the outermost area and outside the harbour, this variation is positive, reaching 2 cm above. Therefore, a difference in level of up to 4 cm can be observed between the inside and outside of the port.

Figure 11 shows the currents in the port of Cartagena and its coastal area according to the results provided by the model for 21 January 2021. Based on the data collected by the measurement campaign, on this day, there was an episode of strong westerly wind, which may have favoured the entry of water into the Escombreras dock at the mouth. According to the results of the model, inflow currents were observed during that day, which would justify the increase in the TR.

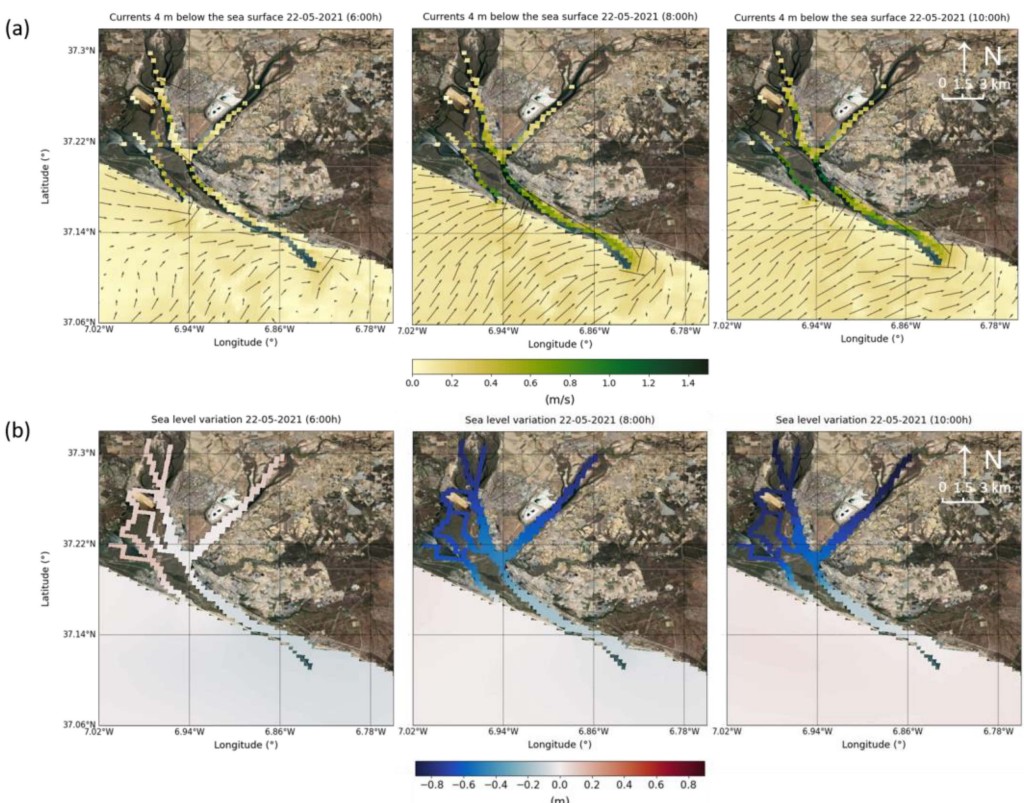

**Figure 8.** Intensity and direction of the sea currents (**a**) and sea level variation (**b**) in the port of Huelva during the episode of increased atmospheric pressure on 22 May 2021.

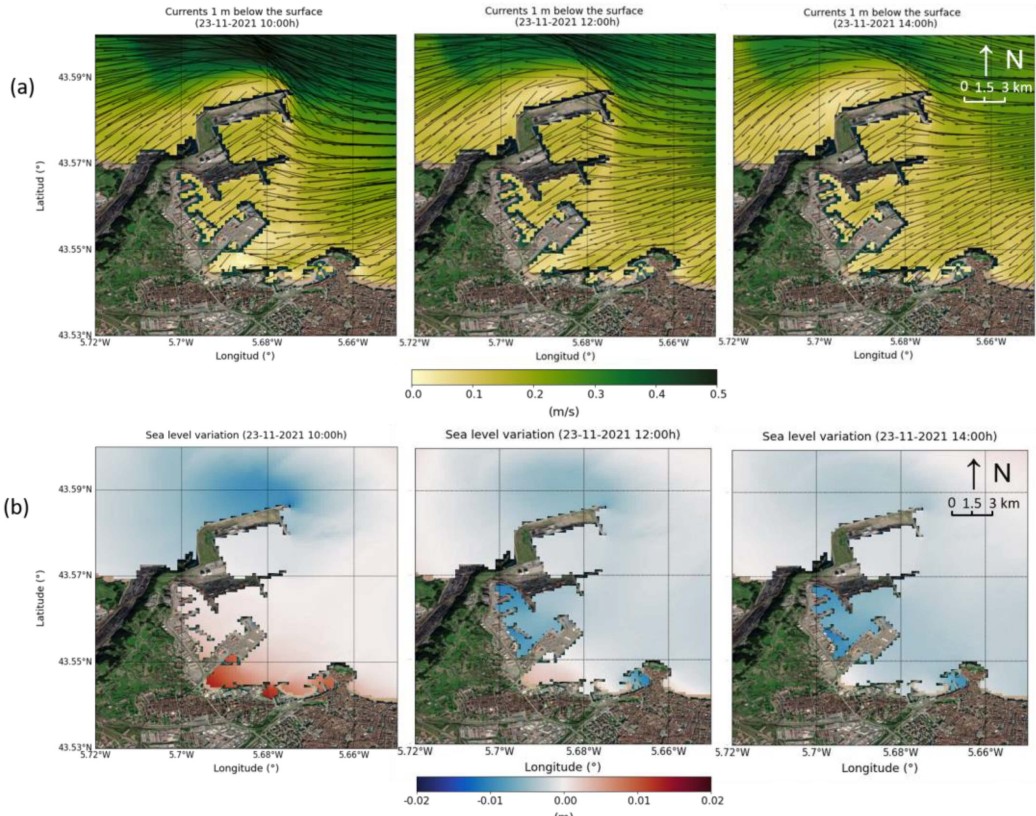

**Figure 9.** Intensity and direction of the sea currents (**a**) and sea level variation (**b**) in the port of Gijón during the northeast wind episode on 23 November 2021.

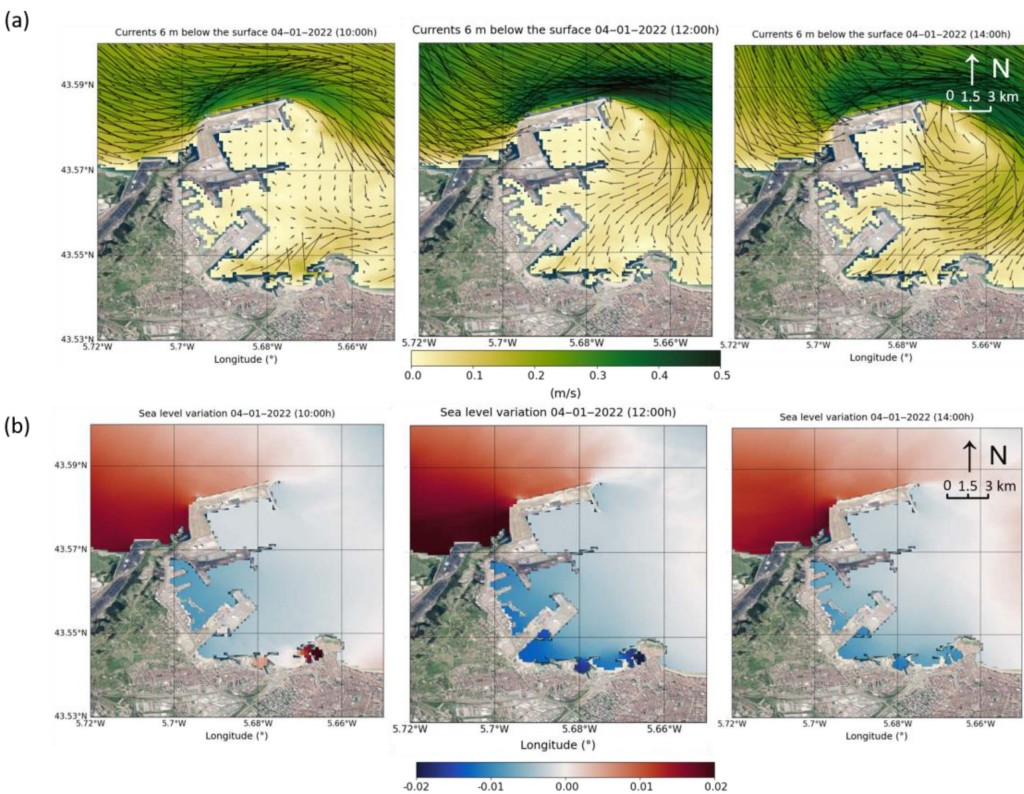

**Figure 10.** Intensity and direction of the sea currents (**a**) and sea level variation (**b**) in the port of Gijón during the episode of increased atmospheric pressure on 4 January 2022.

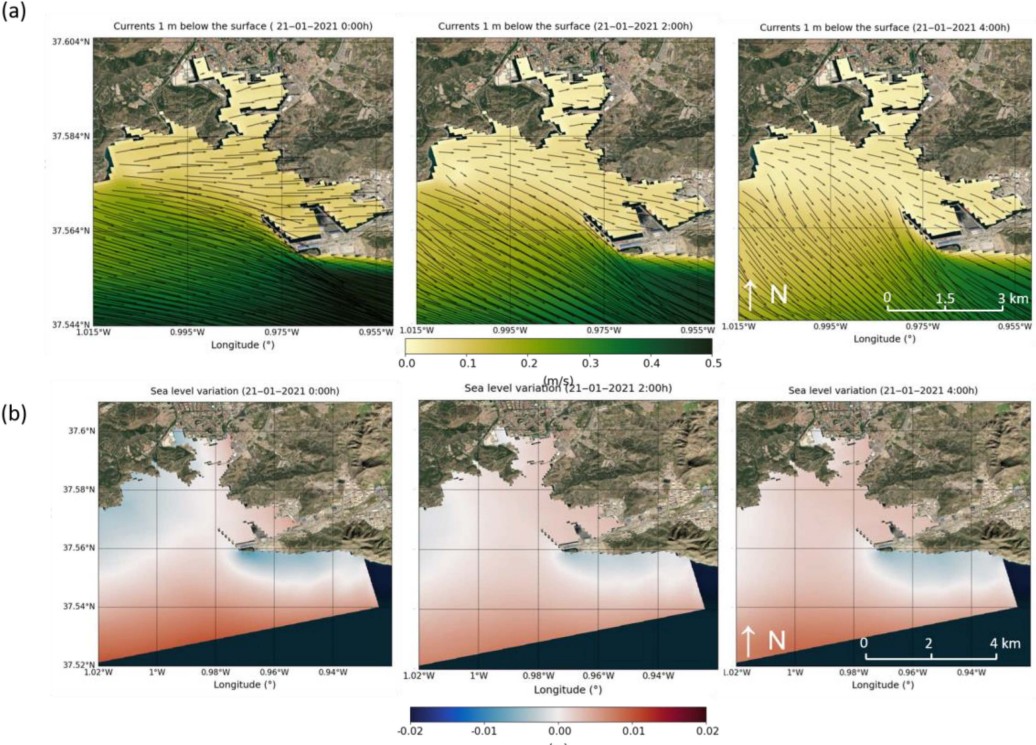

**Figure 11.** Intensity and direction of the sea currents (**a**) and sea level variation (**b**) in the port of Cartagena during the northeast wind episode on 23 November 2021.

Figure 12 shows the results of the model of sea level variation in the port of Cartagena for the day 15 November 2020. In the figure, the sea level variation shows inequalities between the inside and outside of the bay in which the port is located. In the inner zone, this variation is negative, i.e., the level is falling. On the other hand, in the outer zone, this variation is positive, reaching 2 cm above the average level. There is, therefore, a difference in the level of about 3 cm between the inner and outer harbour.

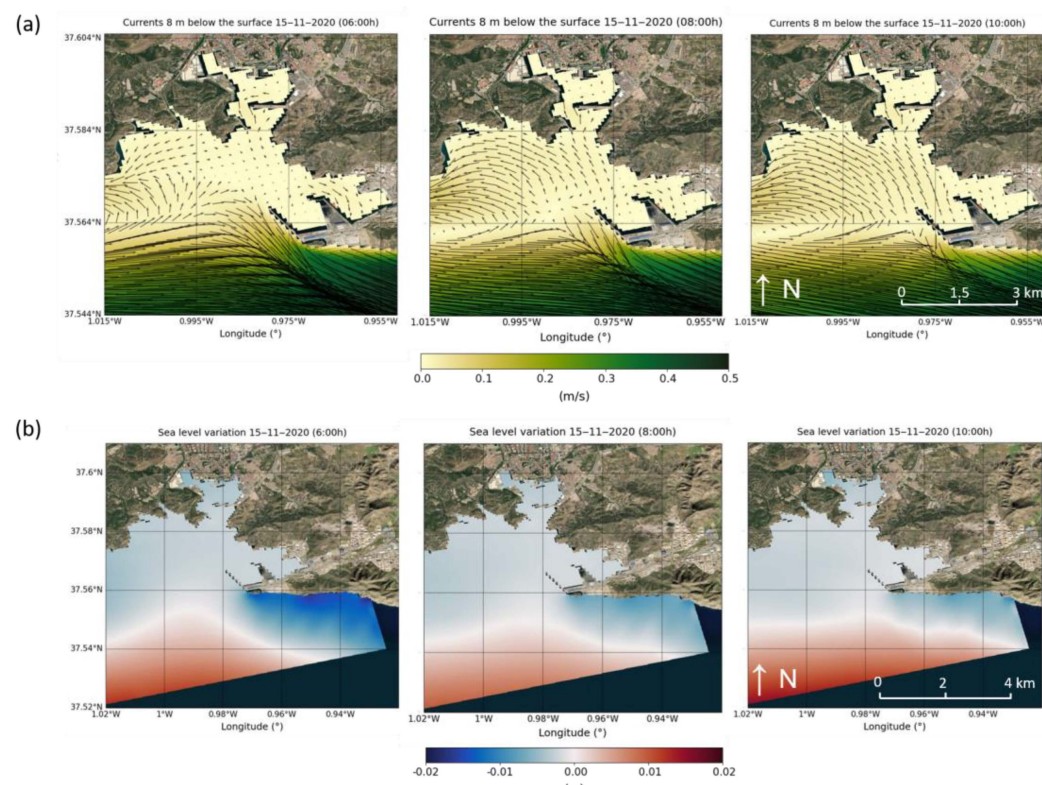

**Figure 12.** Intensity and direction of the sea currents (**a**) and sea level variation (**b**) in the port of Cartagena during the episode of increased atmospheric pressure on 15 November 2020.

## 4. Discussion

Studying the meteorology of harbour regions can be a useful tool for understanding the configuration of water currents, analysing their influence on the interior domain and estimating how they affect the renewal capacity and, therefore, the quality of the water. In this article, the wind and the atmospheric pressure variability effect on the inflow and outflow of Huelva, Gijón and Cartagena harbours have been studied from in situ observations and modelled data.

In the first part of the analysis, the estimated daily renewal time for each of the harbours was compared with the currents measured at the harbour mouth, wind and atmospheric pressure (Figures 4–6). After this first analysis, it was observed that some of the days with high renewal time seem to be related, on the one hand, to wind episodes and, on the other hand, to increases in atmospheric pressure. This coincidence is repeated throughout the time series analysed in the three harbours despite their different physical characteristics.

For the second part of the analysis, six events with high renewal times have been selected, two for each harbour, and the currents and sea level variations have been analysed using data provided by the model (from Figures 7–12). The analysis shows that during these wind episodes, water inflow currents are generated, and during the atmospheric pressure-increase events, a negative sea level gradient between the exterior and interior of the harbour is produced, indicating a lower sea level inside the harbour.

Comparing the information obtained from the analysis of the observations and the model data, it seems that the difference in level between the inner and outer harbour can be generated by the variations in atmospheric pressure (identified previously from the observations) since the volume of water in the harbour is lower than that of the open sea, the sea descends faster and a gap, a difference in level, is produced, which means that water enters the interior of the harbour. It has been observed that this difference in level is not maintained over time but occurs when the atmospheric pressure begins to change.

Figure 13 compares the time series of the level difference (these points are identified in Figure 1) between the outer and inner harbour (data from the model) with the time series of the renewal time (observational data). In this figure, it can be seen that, generally, at times with the highest water level difference, the renewal time is high. This fact could be the origin of an inflow of water when there is a gradient level inside the harbour. Consequently, this water inflow would imply an increase in the water renewal times in the harbour. Therefore, it is probable that, in situations of high atmospheric pressure, inflow currents would be generated to the port, increasing the vulnerability and the risk of water quality problems.

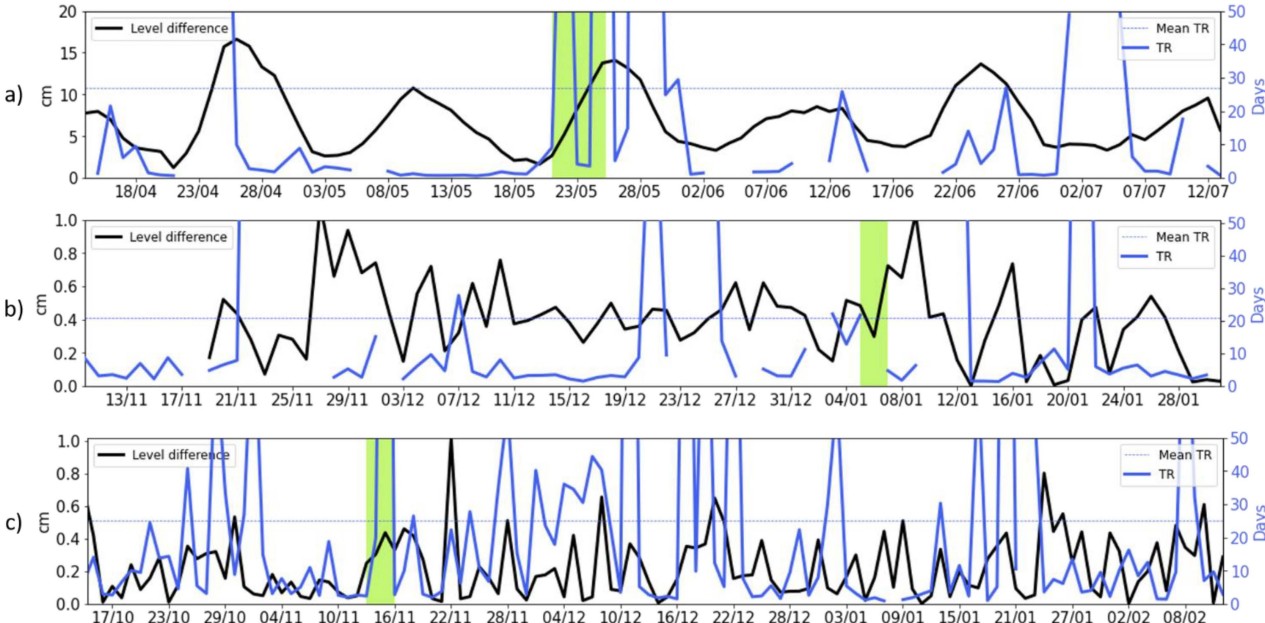

**Figure 13.** Difference (cm) in sea level between the interior and exterior of the harbour (black line) and renewal time (days) in Huelva (**a**), Gijón (**b**) and Cartagena (**c**) harbours. The green boxes highlight the events analysed by the model during which the atmospheric pressure and the renewal time increase. Note the difference of scales on centimetres axes between the different ports for better visualisation.

As shown in Table 6, in the case of the port of Huelva, the proportion of days with TR peaks due to an unknown cause is 48% of the cases. This high percentage may be due to the location of the port in an estuary, where it constantly receives water inflows that were not quantified at the time of the study. These inputs have a significant influence on harbour hydrodynamics and add complexity to the analysis. In addition, due to its channel shape, the wind would drive water inland only from a specific direction.

In the case of Gijón harbour, the main cause of the TR peaks seems to be atmospheric pressure increases (44%) and less probably wind (26%). This may be due, on the one hand, to the fact that the volume of water in this harbour is relatively small and changes in atmospheric pressure have a greater effect. On the other hand, the harbour mouth is sheltered and, therefore, less influenced by the wind.

Finally, in the port of Cartagena, the main cause of the TR increase seems to be the wind (53% of cases). This is due to the fact that, because of its open shape, wind from many directions can blow water into it.

From the analysis, it can be concluded that in the three harbours, most of the TR peaks can be explained by atmospheric forcings (winds or atmospheric pressure). Therefore, it is recommended the use of meteorological stations as a tool for environmental management in harbours and the integration of this information in the hydrodynamic and water quality studies of these domains.

## 5. Conclusions

The main objective of this work was to study how atmospheric forcings can drive water renewal time in three Spanish ports: Huelva, Gijón and Cartagena. These three ports have different geometries and characteristics and are located in different regions of the Spanish coast. However, the results obtained from the different analyses lead to two main common hypotheses for the three ports.

It is observed that during episodes of favourable wind direction at the mouth of the port, inflow currents occur, and TRs increase (for example, on 24 April 2021, 23 November 2021 and 21 January 2021 in Huelva, Gijón and Cartagena, respectively). However, on other occasions, days with high renewal times do not coincide with these wind events. In some of these cases, increases in atmospheric pressure are identified to cause a water level gradient between the inner and outer harbour (e.g., on 25 May 2021, 4 April 2022 and 15 November 2020 in Huelva, Gijón and Cartagena, respectively), generating inflows and increasing the TRs.

The diagram in Figure 14 summarises the second of the hypotheses: changes in atmospheric pressure generate a difference in level between the interior and the exterior of the port (since the volume of water in the interior is less than that of the open sea) which implies the entry of water and, consequently, an increase in the renewal time. In turn, this increase in the renewal time could lead to a worsening of the quality of the harbour water if these conditions are maintained over time.

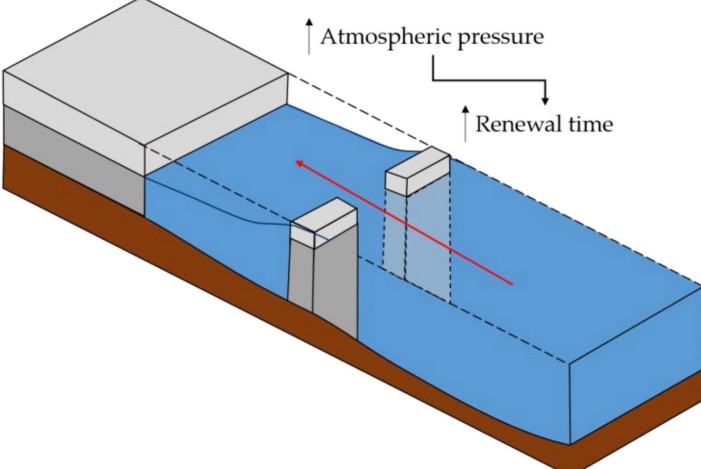

**Figure 14.** Profile of a harbour and its external zone with a water level difference between interior and exterior. The red arrow indicates the inflow at the mouth of the harbour. This figure explains the theory of the movement of water from the exterior to the interior as a consequence of the difference in level generated by increases in atmospheric pressure. The volume of water in the harbour is smaller, and therefore the level decreases faster, which generates a difference in level with the outside and a consequent inflow of water.

The fact that we have estimated renewal times and proposed two hypotheses about their variability based on wind and atmospheric pressure is relevant because these two parameters can be easily measured using basic meteorological stations that are already in general use in harbours. In this way, it would be possible to easily estimate the water exchanges between the port and the open sea, which is essential information for environmental management plans in ports.

The renewal times are usually calculated with data from models due to the lack of observations; however, in this case, data from the measurement campaigns have been used to make these calculations. Studies based only on observations have two main disadvantages. Firstly, the lack of spatial variability when taking data; in this case, only a doppler was available at a specific point; and secondly, the time limitation of the measurement campaign, in this case limited to a few months. Another limitation of this study is the calculation of TR using a single method based on estimated water volumes and outflow rates. The literature report different ways of making this calculation; however, with the available data, it has not been possible to reproduce the calculation with other methods. The installation of doppler equipment and meteorological stations over long periods of time would provide useful data and information to address these deficiencies and difficulties. In this way, it would be possible to confirm and develop in greater detail the hypotheses raised in this study. Finally, only one study has been found regarding the residence time in the Ria de Huelva using a different calculation method [37]. The lack of previous studies has made it difficult to establish a theoretical framework, but it represents a great opportunity to investigate and present novel results on the hydrodynamics, water quality and residence time in the port domains of Huelva, Gijón and Cartagena.

**Supplementary Materials:** The following supporting information can be downloaded at: https://www.mdpi.com/article/10.3390/w15101813/s1, Supplementary Information: Additional information for "Study of Atmospheric Forcing Influence on Harbour Water Renewal". Figure S1: Time series and scatter plots of model results and observations. Figure S2: Histogram of renewal time distribution in the ports of Huelva, Gijón and Cartagena.

**Author Contributions:** Conceptualisation, Y.S., M.E., M.L., M.M., J.M.A. and A.S.-A.; methodology, Y.S. and M.E.; software, Y.S. and M.E.; validation, Y.S., M.E., M.L., M.M. and J.M.A.; formal analysis, Y.S. and M.E.; resources, M.E. and A.S.-A.; data curation, Y.S. and M.E.; writing—original draft preparation, Y.S.; writing—review and editing, Y.S., M.E., M.L., M.M., J.M.A. and A.S.-A.; visualisation, Y.S. and M.E.; supervision, M.E. and A.S.-A.; project administration, M.E. and A.S.-A.; funding acquisition, M.E. and A.S.-A. All authors have read and agreed to the published version of the manuscript.

**Funding:** This research received funding from the EuroSea project, under agreement with the European Social Fund (ESF) through a grant from FI AGAUR 2020 (Agency for the Management of University and Research Grants).

**Data Availability Statement:** Restrictions apply to the availability of these data. Data was obtained from Puertos del Estado. Available online: https://www.puertos.es/es-es (accessed on 5 March 2023) with the permission of this institution.

**Acknowledgments:** This research has received funding from EuroSea project GA862626 funder H2020-EU.3.2.5.1 The authors want to acknowledge the ECO-BAYS research project (PID2020-115924RB-I00, financed by MCIN/AEI/10.13039/501100011033). The lead author has been financed by the Secretaria d'Universitats i Recerca de la Generalitat de Catalunya and the European Social Fund (ESF).

**Conflicts of Interest:** The authors declare no conflict of interest. The funders had no role in the design of the study; in the collection, analyses, or interpretation of data; in the writing of the manuscript; or in the decision to publish the results.

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
