# Peer review of "Study of Atmospheric Forcing Influence on Harbour Water Renewal"

_water, doi:10.3390/w15101813_

Round 1

Reviewer 1 Report

The authors have examined the influence of atmospheric forcings on water renewal rates in harbour domains. Specifically, to examine the effect of weather conditions on water inflows and outflows in ports based on observations and modelled data. In other words, to relate the variability of the renewal time of each harbour to meteorological weather, in particular wind and atmospheric pressure. The concept is interesting and nicely presented. However, the following points should be addressed before its acceptance for the publication.

 1.     Abstract must provide precise information of the key findings of the present work. I suggest revising the abstract to highlight the novelty of the work.

 2.     Introduction: I suggest the authors expand the introduction, to include recent studies and to provide enough background and explanations to make the paper informative for a wider audience. The authors should highlight the novelty of the present work, and describe the difference with others.

 3.     What is the significance of the SAMOA forecasting system?

 4.     The authors should cite some recent papers published in Water, to show that this study is within the scope of the journal.

5.     Provide the detailed description of implemented numerical technique used for the numerical simulation. 

 6.     It is also suggested to provide the used hydrodynamical mathematical models.

 7.     Check the accuracy of the references.

Overall, the quality of the paper is good, and I recommend for its publication after minor corrections as suggested. 

Reviewer 2 Report

The paper presents a new and interesting tool to predict weather effects on currents and water levels in ports and the presented case studies cover different port types and tidal ranges. However, the material and methods part must be improved. There is no detailed description on how the renewal time is computed. The validation process should be better explained and the low correlation values presented in table 4 should be discussed as well as strategies to improve them. In the results section, table 5 is almost impossible to understand. The first 3 paragraphs of the discussion section seem superfluous. However, the argument around figure 13 is very interesting.

In the attached pdf file, there are some comments and corrections that can be seen with the Adobe Acrobat reader.

Author Response

Dear reviewer, many thanks for spending your time to review this article. It is important for us to know your opinion and to apply your recommendations. Your suggestions are answered point by point below.

The paper presents a new and interesting tool to predict weather effects on currents and water levels in ports and the presented case studies cover different port types and tidal ranges. However, the material and methods part must be improved.

  1. There is no detailed description on how the renewal time is computed.

The explanation of the renewal time calculation has been expanded with a new section (2.3 Renewal times) in the manuscript.

  1. The validation process should be better explained and the low correlation values presented in table 4 should be discussed as well as strategies to improve them.

The explanation of the validation process has been extended and bibliographical references have been added. It is true that the correlation coefficients between model results and observations are not excellent, so the model results are only used qualitatively. That is, they do not accurately reproduce current intensities or the number of centimetres of sea level variation, but they do reproduce trends. If the observations show inflow currents or sea level rises, the model (although with slightly different values) does reproduce these inflow currents or rises.  

  1. In the results section, table 5 is almost impossible to understand.

The footnote has been rewritten and an explanation has been added to the text above: Table 5 shows the average renewal time calculated from the currents measure-ments at each harbour mouth. These results are 27, 21 and 25 days for the ports of Huelva, Gijón and Cartagena respectively. The days above average are 28%, 27% and 25% of the total series in each case. In order to understand the origin of these above-average TR values, wind and atmospheric pressure variations (from campaign data), together with currents and sea level (from observations and model results) are studied.

  1. The first 3 paragraphs of the discussion section seem superfluous.

The first two paragraphs of the discussion have been deleted.

  1. In the attached pdf file, there are some comments and corrections that can be seen with the Adobe Acrobat reader.

Comments have been answered in the same PDF, please find it attached.

The new manuscript will be uploaded when the revisions of the other reviewers are completed. 

Reviewer 3 Report

The aim of this study is explicitly stated in lines 98-99 as "to analyse the influence of atmospheric forcings on water renewal rates in harbour domains", using observations and numerical simulations. Thus, the conclusions of the study should be based on a robust statistical handling of data, mainly with regard to correlations, and then an effort to support "causality" over "casualty" (in the archaic sense of chance). I did not find either.

1. For starters, the choice of the graphs in Fig. 3 look suspiciously as "cherry picking"; why only a different one for each location? The three graphs should be shown for the three locations. 

2. The correlation coefficients "R" between model results and observations are good for atmospheric pressure, but extremely low for almost everything else, exception made of "meteorological tide" for Cartagena and "currents through the mouth" for Huelva, which happen to be the data in Fig. 3! Though the authors admit that "(g)eometric complexity of each domain and tides intensities can have a significant influence on measurement and predictions", this part should be discussed with more detail, especially considering that "R" is meaningful for linear relationships only. While the graphs in Fig. 3 provide information about the behavior of the variables as a function of time, the graphs with the model's data compared to the observations should be provided as supplementary information, and the straight line y = x drawn in all of them. 

3. The numerical data in Table 5 do not suffice to show the behavior of the currents. The respective histograms would be much better, especially with regard to the possible asymmetry of the distributions. 

4. I do not understand the vertical scale of graphs 4a, 5a and 6a. How is the TR calculated?

5. What are the blue dots in graphs 4d, 5d, 6d?

6. How did the authors arrive at the numbers in Table 6? That is, how can they assure what is the cause of renewal times above average? The comparison of the corresponding histograms of days with certain TR to those of atmospheric pressure and wind speed would help to make that clear. That is, besides explaining it in the text.

In view of all the concerns I have, I cannot recommend publication of the typescript.

Reviewer 4 Report

- The subject addressed is within the scope of the journal.

- The abstract could become much better if it properly introduces the study from a research standpoint. Also, the main findings could be stated more pointedly in the abstract.

- However, the manuscript, in its present form, contains several weaknesses. Appropriate revisions to the following points should be undertaken in order to justify recommendation for publication.

- All symbols and parameters should be defined, please check.

- Please add a proper legend for the Figure 1.

- Some key parameters are not mentioned. The rationale on the choice of the particular set of parameters should be explained with more details. Have the authors experimented with other sets of values? What are the sensitivities of these parameters on the results?

- It is suggested to add articles entitled “Sazonov et al. Simulation of Hybrid Mesh Turbomachinery using CFD and Additive Technologies” and “Amr Ismaiel. Wind Turbine Blade Dynamics Simulation under the Effect of Atmospheric Turbulence to the literature review.

- Conclusion:

•The conclusion section is currently a repeat or rehash of the preceding sections, and  needs to be re-written to improve it, keeping in mind the following suggestions.

•Update the conclusion to include the newly formulated theoretical contributions

•Mention the limitations of the study and prospects for future research.

•Summarize the key results in a compact form and re-emphasize their significance.

•This conclusion could be worded in a manner as to emphatically motivate the academic community to get down to actionable, practical engaged scholarship.

- Page 5: the following paragraph is unclear, so please reorganize that:

“Table 5 shows the renewal times information obtained from the current measurements at the mouth of each port. Average values of 27, 21 and 25 days are observed for the ports of Huelva, Gijón and Cartagena respectively. The days above average are 28%, 27% and 25% of the total series in each case. In order to understand the origin of these results, wind and atmospheric pressure variations (from campaign data), together with currents and sea level (from observations and model results) are studied.”.

Author Response

The new manuscript will be uploaded when the revisions of the other reviewers are completed. 

Reviewer 5 Report

In this manuscript (“Study of atmospheric forcing influence on harbour water renewal”, authorships of Yaiza Samper, Manuel Espino, Maria Liste, Marc Mestres, Jose Alsina and Agustin Sánchez-Arcilla), which was submitted to journal "Water" (MDPI), authors investigate kinematics of atmospheric forcing influence on harbour water renewal on sufficiently good level with aim of predicting the further water balance and renewal  inside the chosen (Spanish) harbours. They demonstrate the good level of understanding with respect to such kinematic mechanism in this or that harbour.

General ansatz seems to be professionally presented, all the simple logic and kinematic manipulations are under responsibility of the authors.

My only recommendation is to slightly update the level of scientific consideration (I mean overall conception and methodology) of this study by mentioning research regarding influencing of exisiting Large-Scale Coherent Structures in the motions of water in the Ocean near the coastal regions and inside the basins of this or that chosen harbour:

1. Ershkov, S.V.; Rachinskaya, A.; Prosviryakov, E.Y.; Shamin, R.V. On the Semi-Analytical Solutions in Hydrodynamics of Ideal Fluid Flows Governed by Large-Scale Coherent Structures of Spiral-Type. Symmetry 2021, 13, 2307. https://doi.org/10.3390/sym1312230;

2. Samelson, R.M. Lagrangian Motion, Coherent Structures, and Lines of Persistent Material Strain. Annu. Rev. Mar. Sci. 2013, 5, 137–163.
3. Gnosh, A.; Suara, K.; Yu, Y.; Zhang, H.; Brown, R.J. Using Lagrangian Coherent Structures to Investigate Tidal Transport Barriers in Moreton Bay, Queensland. In Proceedings of the 21st Australasian Fluid Mechanics Conference Adelaide, Adelaide, Australia, 10–13 December 2018.

4. Ershkov, S.V.; Shamin, R.V. A Riccati-type solution of 3D Euler equations for incompressible flow. J. King Saud Univ.-Sci. 2020, 32, 125–130.

5. Verma, V.; Sarkar, S. Lagrangian three-dimensional transport and dispersion by submesoscale currents at an upper-ocean front. Ocean Model. 2021, 165, 101844.
6. Zhang, Y.-W.; Feng, Y.-L.; Feng, C.-A.; Wang, Z.-W.; Zhang, X.-Q. Study on Lagrangian Coherent Structure of tidal current field in Laizhou Bay. Shuidonglixue Yanjiu yu Jinzhan/Chin. J. Hydrodyn. Ser. A 2021, 36, 95–101

Nevertheless, I should especially note that the presented manuscript is the self-consistent development in such area of researches in environmental hydrodynamics. My recommendation: minor revision. I wish to review this article after revision again.

Author Response

(The authors gave the same response as above.)

Reviewer 6 Report

Comments for “Study of atmospheric forcing influence on harbour water re- 2 newal

It is an effective method to the effect of meteorological parameters such as wind and atmospheric pressure on harbour water exchanges using SAMOA forecasting system. This study still needs to be improved before being published.    

1、The introduction is too scattered. It is recommended to organize it into three paragraphs.

2、The profile map of the study area is very non-standard, with no latitude or longitude.

3、Please further refine the innovation of the research in introduction 

4、Whether wind speed or wind direction is considered in the model simulation, and whether the effect of wind direction is quantified

5、How representative is the date selected in figure 7-12

6、The model verification results are given in Table 4, but the verification results of vertical simulation are not given

7、How is model parameter localization handled

The paper is the application of the model. The study is interesting, but the writing is too poor and the degree of conciseness is insufficient, especially the introduction and conclusion

Reviewer 7 Report

The manuscript investigated the influence of atmospheric forcings on water renewal rates in harbour domains. It examined the effect of weather conditions on water inflows and outflows in ports based on observations and modelled data. I consider the content of this manuscript will meet the reading interests of the readers of the journal. Some interesting and valuable results were obtained and the results are meaningful. Overall, the manuscript is carefully prepared and well written.

Author Response

(The authors gave the same response as above.)

Round 2

Reviewer 3 Report

The authors addressed my queries satisfactorily; nonetheless, I suggest to explain what is the red straight line in the caption of Fig. S1. I understand the difficulty of trying to find causality in a complex, multi-factorial phenomenon as the mouth current intensities, as shown by graphs in Fig. S2. It would be worth to attempt an explanation of the much better fit in the Huelva harbour. However, I don't object publication in the journal of the typescript as it currently is.

Reviewer 4 Report

The article has been revised very well, so I would suggest to accept in its present form.